# 'Was my kidney biopsy worth it?'–A qualitative phenomenological study of patient experiences and perceived barriers to kidney biopsy

**Michael Toal** [1]*, **Megan Raynor**[2], **Clare McKeaveney**[2], **Ciaran O'Neill**[1], **Michael Quinn**[1], **Christopher Hill**[3], **Alexander Peter Maxwell**[1]

**1** Centre for Public Health, Queen's University Belfast, Belfast, Northern Ireland, **2** School of Nursing and Midwifery, Medical Biology Centre, Queen's University Belfast, Belfast, Northern Ireland, **3** Regional Nephrology and Transplant Unit, Belfast City Hospital, Belfast, Northern Ireland

* mtoal11@qub.ac.uk

## Abstract

### Background

Kidney biopsy is an important investigation in nephrology and facilitates the diagnosis of many conditions. It is an invasive procedure with the risk of significant complications, which limits its usage. There is minimal literature on how patients experience a kidney biopsy. Identifying and addressing barriers to access may expand opportunities for diagnosis and treatment. We hypothesise that patients experience kidney biopsy differently, depending on each individual's circumstances.

### Methods

Ten participants, who had undergone a total of twenty-three kidney biopsies were recruited through purposive sampling. They were interviewed about how they experienced the procedure, how they felt it had impacted their own medical care and about potential barriers and facilitators to access for other patients. A descriptive phenomenological approach was utilised and thematic analysis was applied to responses.

### Results

Three main themes emerged: Unforeseen health concerns discovered, resilience and re-evaluation and the need for a patient-centred approach to biopsy. The experience of pain and discomfort varied amongst patients, but there was a significant emotional and psychological toll associated with kidney biopsy. All patients felt that the procedure had a positive impact on their treatment course through increased diagnostic information for them and their healthcare team. Further information in advance and the presence of trusted healthcare staff were identified as facilitators to kidney biopsy.

**Data Availability Statement:** Data cannot be shared publicly as the detailed qualitative data

could be potentially disclosive if full transcripts were available to the public. The approving research ethics committee did not permit adding this data to a public repository and participants were not consented to have their data freely available. However, certain data may be available upon reasonable request by contacting Lorraine Carew, Research Ethics Officer, Faculty of Medicine, Health and Life Sciences, Queen's University Belfast at facultyrecmhls@qub.ac.uk.

**Funding:** MT is supported by a research award from the Northern Ireland Kidney Research Fund. This organisation was not involved in the design or conduct of this study. https://nikidneyresearch.org/ .

**Competing interests:** The authors have declared that no competing interests exist.

## Conclusion

Kidney biopsy is experienced differently by patients. Improved information in advance by trusted healthcare professionals may reduce patient-related barriers to biopsy access.

## Introduction

Kidney biopsy (KB) is an important investigation in nephrology and is the only way to definitively diagnose many renal disorders [1]. This is a routine procedure within nephrology units, with almost 2500 native kidney biopsies performed in Scotland over a five-year period and international native biopsy rates range from 10–230 procedures per million population per year [2]. KB is performed either as a day case or during an inpatient episode. Following the procedure, patients are normally asked to remain in bed for several hours of observations [3].

KB is an invasive procedure and as such is associated with potential complications. The most common complications relate to bleeding due to the highly vascular structure of the kidney. Most complications are self-limiting and do not result in significant adverse outcomes, but major complications can occur [3]. A perinephric haematoma has been observed on CT or ultrasound imaging in 57–91% of biopsies, however most individuals are asymptomatic [4, 5]. Macroscopic haematuria has been reported to occur in around 3% of KBs and 1% of patients require a blood transfusion. Major bleeding complications requiring radiological embolisation or nephrectomy are rare, occurring in 0.3% of biopsies. Death related to KB is exceptionally rare at 0.06% [6].

There is very limited reported literature exploring how patients experience undergoing this invasive procedure. A US study reported on 111 participants, 28 days after a protocol KB, and found that 63% of patients reported pain after the biopsy, but mostly below 4 on a 0–10 scale. 64% reported anxiety before biopsy, falling to 9% following the procedure [7]. A Danish qualitative study followed seven patients over a seven-hour period during their entire admission for KB [8]. These researchers described three main themes: patients required their basic needs to be met, they needed information and reassurance, and humour was used as a coping strategy [8].

This study aims to explore how patients experienced KB, how they perceived undergoing this procedure had impacted on their long-term care and what barriers and enabling factors could be identified to improve access to KB for other patients.

## Materials and methods

Findings are reported in line with the consolidated criteria for reporting qualitative research (COREQ) guidelines [9]. Interviews were conducted by MT, a male specialist trainee doctor in nephrology. MT has trained in qualitative research methodologies through postgraduate education and has also performed many kidney biopsies in his clinical work, however he was not involved in the clinical care of any study participants. A participant information sheet (PIS) was produced and distributed to potential participants explaining the role of the interviewer, purpose of study and reassurance that their participation would remain confidential and have no impact on their medical care. Prior assumptions held by the interviewer were that patients may be reluctant to undergo an invasive procedure if they had minimal symptoms and the fear of pain acted as a barrier to biopsy.(10) A short explanatory video was produced in conjunction with the university media department to outline the rationale for the study, which included contact details for MT and the patient representative (MR). The video was

disseminated through local and national kidney patient groups via email and social media out-lets including: WhatsApp, Facebook, Instagram and Twitter/X.

Purposive sampling was used. Participants suitable for inclusion were adults over the age of 18 years at the time of interview, had self-reported to have had a KB under the care of a nephrologist, proficient English language skills to facilitate an interview and were able to give consent to participate. Individuals who came forward were sent a PIS in advance, then invited for a virtual interview which was conducted and recorded through Microsoft (MS) Teams or phone. Verbal consent was recorded before every interview and no contemporaneous field notes were taken.

The format of the interview was semi-structured (S1 Fig) and focused on the experience of kidney biopsy and reflections on how it has impacted care. Interviews were held from 20th March- 25th April 2023 with duration ranging from 9 to 32 minutes. Transcripts were not returned to participants before or after analysis and were generated using MS Teams software. The pre-specified criteria for data saturation were two consecutive interviews which yielded no new unique themes or subthemes, which was met after ten interviews had taken place.

A qualitative approach for this study involved phenomenology which aims to describe the lived experience of everyday phenomena as well as understand how people attribute meaning to that experience [10]. Neubauer et al explain that phenomenology "is to describe the mean-ing of this experience—both in terms of *what* was experienced and *how* it was experienced" [11]. Reflected in the current study, a phenomenology approach was used to understand the lived experiences of kidney biopsy and derive meaning unique to each individual [12–14]. Complementary to this qualitative approach, Colaizzi's descriptive phenomenological analysis was utilised to derive the lived experience using several steps. Following and in agreement with several authors, this was completed by step 6 [15, 16]. Firstly, the researchers studied the tran-scripts in detail to familiarise themselves with the findings and identify significant statements which could be used to formulate meaning, whilst using 'bracketing' to try to separate their own biases from influencing interpretation. Themes were then clustered into common experi-ences across all participants. These results were used to write a full description of the phenom-enon in detail. The analysis was completed by condensing the description into succinct themes and subthemes that capture the essence of the phenomenon [14].

Coding was contemporaneous after each interview. Themes and subthemes were derived from transcripts by the interviewer (MT) and then transcripts were independently coded by an experienced qualitative researcher (CMcK). The final themes and subthemes were agreed following several meetings and discussions between MT and CMcK, as well as the wider research team.

Ethical approval for this project was granted by the Faculty of Medicine, Health and Life Sciences (FMHLS) Research Ethics Committee (REC) of Queen's University Belfast (Project no. MHLS 22_175) on 15th February 2023. Research was conducted in accordance with the Helsinki declaration.

## Results

### Participants

Ten participants were recruited for the study, with no withdrawals. Baseline characteristics are described in Table 1. Six participants identified as female and four as male. Age at the time of interview ranged from 29–56 years. Age at the time of biopsy ranged from 14–47 years. Two participants lived in England, eight lived in Northern Ireland. One participant reported his ethnicity as British-Pakistani, all other participants identified as White. These ten individuals had undergone twenty-three kidney biopsies: six native and seventeen transplant biopsies.

**Table 1. Baseline characteristics of interview participants.** RRT = renal replacement therapy. *Age unknown for two of four biopsies.

| Identifier | Age at interview | Sex | Ethnicity | No. of biopsies | Age at biopsy | Biopsy subtype | Current RRT |
|---|---|---|---|---|---|---|---|
| P1 | 47 | Female | White | 1 | 29 | Native | Transplant |
| P2 | 38 | Male | White | 2 | 34 | Native | Transplant |
| P3 | 41 | Female | White | 2 | 23, 35 | Native & Transplant | Transplant |
| P4 | 43 | Female | White | 2 | 32, 38 | Transplant | Transplant |
| P5 | 37 | Male | British-Pakistani | 4 | 14*, 36 | Transplant | Transplant |
| P6 | 56 | Male | White | 3 | 28, 47 | Transplant | Transplant |
| P7 | 29 | Female | White | 2 | 17, 20 | Transplant | Transplant |
| P8 | 30 | Female | White | 2 | 15, 21 | Native & Transplant | Transplant |
| P9 | 56 | Female | White | 1 | 47 | Native | None |
| P10 | 46 | Male | White | 4 | 31, 34 | Native & Transplant | Transplant |

Nine participants had undergone a kidney transplant and one participant had chronic kidney disease.

## Main themes and sub-themes

Three main themes were identified: unforeseen health concerns discovered, resilience and re-evaluation and a patient-centred approach to kidney biopsy. Themes and subthemes are summarised in Table 2. Findings emphasised the physical and emotional impact of unexpected health concerns, the significance of coping and the consequences, and the importance of aligning treatment with the patient's goals and values.

## Theme 1: Unforeseen health concerns discovered

**Sub-theme 1a: Deceptive health condition.** The early stages of kidney disease are associated with subtle symptoms or can be entirely asymptomatic. For this reason, patients may be unaware of their condition until it proceeds to an advanced stage.

**Table 2. Map of themes and subthemes emerging from interviews.**

| Themes | Subthemes | Explanation |
|---|---|---|
| Unforeseen health concerns discovered | Deceptive health condition | Participants reported minimal physical symptoms when abnormalities were detected |
| | Unexpected emotional elements of biopsy | The procedure invoked strong emotional reactions during and after the biopsy |
| | Physical discomfort and monotony | Participants described significant pain associated with the procedure and boredom during the long observation period |
| Resilience and re-evaluation | Need for resilience | Resilience of participants was tested due to the challenging circumstances that arose |
| | Re-evaluation of priorities | The period of illness and investigation prompted participants to reconsider their goals and responsibilities |
| Patient-centred approach to kidney biopsy | Trusting staff | Participants report feeling at ease when staff performing the biopsy were known and trusted by them already |
| | Improved information provision | Participants felt that further information about the process and potential results in advance of the biopsy would have alleviated their apprehension |
| | Psychological support | Some participants found their illness and medical care took an emotional toll and further psychological support may be warranted. |

"(For) six months I was cramping after football on Friday night."–P2

Over this period, some participants remarked that they were still able to function well and continue their normal activities, which were often of high intensity. Some found it difficult to understand how their kidney function could be so poor when they felt reasonably well.

"I was diagnosed, I had like 15% function left, which is on the one hand amazing that I was still standing and cycling"–P10

**Sub-theme 1b: Unexpected emotional elements of biopsy.** Attending for a KB was an emotionally challenging experience for many participants. Often, they were dealing with a new diagnosis of a serious medical condition and being asked to undergo an invasive procedure within a short space of time.

"Within 24 hours (from admission), they were saying that it's looking like I need a transplant"–P2

"The sound of the biopsy was like a gun. I think that was the worst bit about it."–P3

Difficulty with urination was noted by some female participants, as the strict bed rest made this challenging. The loss of autonomy and reliance on health professionals to fulfil basic needs placed a considerable emotional burden.

"It was a little bit humiliating, having nurses bring a bedpan."–P1

"(Using a bed pan) was the most undignified experience I've ever had"–P3

**Subtheme 1c: Physical discomfort and monotony.** Participants experienced varying levels of physical discomfort during the KB. Some individuals reported severe pain both during the procedure and afterwards for several days.

"It's (the) most painful thing I've ever felt in my life."–P2

In contrast, many participants reported that they experienced minimal pain or discomfort associated with their kidney biopsy.

"I wasn't in pain at all"–P3

After a kidney biopsy, patients are observed for 4 to 6 hours, which some participants found monotonous.

"You just keep on looking at the clock and it's almost like the clock is stopped"–P6

Participants who had multiple kidney biopsies describe having a more positive experience of this aspect on subsequent procedures, as they came prepared with ways to occupy the time.

"I was more prepared (for the second biopsy). I brought more things with me in terms of doing a Sudoku or doing different bits and pieces"–P3

### Theme 2: Resilience and re-evaluation

**Sub-theme 2a: Need for resilience.** Resilience was required not only for undergoing the procedure itself, but also for the consequent diagnosis and treatment. Participants demonstrated resilience to continue to live their lives to the full, despite these setbacks.

"I think I use humour to deflect from trying to remember everything and think about everything. But yeah, it's had its ups and downs let's say."–P5

Some participants adopted a pragmatic approach and had the understanding that a kidney biopsy was a necessary step towards receiving the optimal medical care.

"Kidney biopsies are part and parcel of the process. . .it has to be done, so there's no point stressing or worrying about it"–P4

**Sub-theme 2b: Re-evaluation of priorities.** Participants described a change in their attitude towards other people and life goals. Some felt a sudden change not only within their own lives, but also to others around them, placing a strain on established relationships.

"It's opened my eyes to that bigger picture of, for me, who was there?"–P2

"My wife was very good, but she sort of kept me in the dark about certain things. You know, the kids were having trouble in school, which I didn't learn until a year and a half later"–P2

One participant described a loss of identity and role, as he was no longer able to participate in activities he enjoyed, therefore losing social connections.

"When my kidneys failed when I was 28 years old that was the biggest thing that I missed was the ability to do rugby and the social aspect. . . I got a lot of satisfaction of playing the game and suddenly that was just taken away from me."–P6

### Theme 3: Patient-centred approach to kidney biopsy

**Sub-theme 3a: Trusting staff.** Many participants placed an enormous amount of trust in their medical team and highly valued the care they had received over a long period of time.

"I'm very lucky, I have great consultants, so they phoned me as soon as the results came in. . ... I had a lot of confidence in the people who were doing (the biopsies), so I suppose that helps."–P4

Some participants reported that their anxiety surrounding the procedure was alleviated by the presence of healthcare staff that were known to and trusted by them.

"I think it was two biopsies performed by two different consultants. But I had complete trust in both"–P6

Participants valued the diagnostic certainty obtained from KB, both for themselves and their healthcare team, which helped guide their treatment.

"I think this is probably one of the best ways to really get under the hood and to see whether, you know, the kidney is really performing well. So I guess you can check your

bloods and you can check all kinds of other things. But this is really getting into the machine itself. . .it gave me a lot of comfort knowing, for example, when I had that deterioration of function momentarily that I knew that there was nothing wrong"–P10

**Sub-theme 3b: Improved information provision.** Participants acknowledged that there were barriers and challenges associated with KB that were difficult for them or could be for patients in a similar position.

"I guess fear. I guess not fully understanding why it's needed. What might be involved? How long it takes, but also how long afterwards it takes. What potential results could come back?"–P5

Participants reported anxiety associated with unknown elements of attending for the KB.

"(Advice for future patients)- I would say this is what's gonna happen. This is what the machine looks like. It's gonna be bloody scary. It's a loud noise, prepare yourself for a loud noise. And over and above the things that you're normally told, Prepare yourself for an extremely long wait."—P3

**Sub-theme 3c: Psychological support.** Some participants found the process of having a kidney biopsy, receiving a diagnosis and undergoing treatment overwhelming.

"My head wasn't right. I found myself challenging everything"–P2

"I was in school and like, I was so stressed about, like keeping up with my peers,"–P7

Despite attempts to maintain normalcy through denial, the challenge of keeping pace with peers and confronting the reality of their condition ultimately brought about mental distress and internal conflict.

"So when (Doctor) left (after explaining the biopsy results), my mum came in and she had been crying. And so that set me off crying. And then we just sort of as a family sat and, like processed what was gonna happen. . ..I was in denial. I thought there was nothing wrong with me. And then this was the actual final bit that confirmed well actually, this is going to be longer than I thought I would be in for"–P8

Participants also explored the facilitators that would lessen the burden and anxiety for patients coming forward for biopsy. The beneficial effect of social connections was revealed, as participants reported that having trusted staff around them or peer support from other patients could help overcome the barriers to access.

"I'd like to see much more kind of recognition of and acknowledgement of patient empowerment of their own mental health and wellbeing. . . I'm always kind of one that hammers on about peer support and getting patient peer support involved"–P6

## Discussion

Participants varied in their experience of a KB. Participants reported differing degrees of pain associated with the biopsy procedure which appeared to be unrelated to sex or age. There did not appear to be any obvious associations to explain this discrepancy and perceptions of pain

varied widely in this small group of participants, none of whom had a major complication of KB. The link between psychological distress and pain perception is well established and higher levels of pre-operative emotional distress has been shown to predict increased opioid use in the post-operative period [17].

Participants who had undergone a kidney transplant before their first biopsy appeared to better tolerate the procedure. There may be several reasons for this including the previous experience of an invasive procedure, a longer relationship with their healthcare team and a better awareness of the clinical environment. Additionally, the supine positioning for a transplant biopsy (rather than prone for native biopsy) could improve communication with the operator. However, contrasting the experience of native and transplant biopsies was not the objective of this study, therefore caution is warranted in interpretation.

Other factors like age and sex appeared less influential in experiences of a KB, however this is difficult to determine in a small study population. Rehearsing the processes required for kidney biopsy in the form of interactive resources or guided imagery may help alleviate anxiety, as participants reported that subsequent biopsies were generally better tolerated than their first experience.

The observation period after KB was challenging for individuals who were not equipped with resources to pass the time, but other participants used this time effectively. No major complications associated with KB occurred within this cohort.

The emotional and psychological toll of living with kidney disease should not be underestimated, which often persists despite successful transplantation [18]. Although the focus of this study was the impact of kidney biopsy, some participants described in detail their experience of living with kidney disease. Although this may not always be directly related to the biopsy, it is inevitably intertwined with their lived experience, as the biopsy occurred in the context of living with their disease. The participants who volunteered for this study appeared to be motivated to inform and support other patients by sharing their experience, as peer education is valued by patients living with kidney disease [19].

## Limitations

This study has notable limitations. Transferability is limited, as the experience of only ten individuals is captured, who were often affiliated with patient groups and may have comparatively greater healthcare engagement. Recall bias may also influence responses, as there were many years between KB and interview for all participants. The interviewer (MT) from within the medical profession may have influenced how participants responded to questions. Opportunistic sampling resulted in some participants coming forward who knew MT through local patient groups, however he was not directly involved in the care of any participants.

## Strengths

This study also has significant strengths. It acts as a solution to the main barrier reported by participants- a lack of information in advance. This study examines the process of KB from the patient's perspective and should act as reassurance, as participants felt that transient discomfort has resulted in improved care. International rates of KB are highly variable [2]. By understanding patient-related barriers and enablers, healthcare providers should aim to mitigate circumstances that make it difficult for patients to come forward for biopsy, such as addressing limited healthcare literacy and adjusting for conflicting roles such as full-time carers or self-employed persons, to help improve equitable access to diagnosis and treatment of kidney disease.

## Implications for future research

Larger and more diverse participant numbers are needed to capture unique circumstances, such as the effect of biopsy complications and how perception is influenced by life circumstances. A patient-centred approach to research offers the opportunity for patient stakeholders to guide research priorities to ensure that their needs are at the forefront of new developments. Kidney patient charities do have online resources for patients undergoing KB, however patients may not be aware of these and signposting by health professionals may help expand their usage [20, 21].

## Conclusion

Participants who had undergone a KB reported varying degrees of physical and emotional distress, however the diagnostic information obtained was felt to have positively impacted their treatment course. Trust in healthcare professionals and accessible information in advance of the procedure, as well as guidance through the results and consequent treatment was highly valued. Further patient-centred research is needed to identify and remove systemic barriers to healthcare access.

## Supporting information

**S1 Fig. Semi-structured interview guide.**
(TIF)

## Acknowledgments

The authors would like to thank the following patient groups for sharing information on this study through email and social media: Northern Ireland Kidney Patients Association, Transplant Sport Northern Ireland, Kidney Care UK and Young Adults Kidney Care Group UK. The promotional video was produced by Stephen Mullan, Video Resources Producer, Queen's University Belfast (QUB).

## Author Contributions

**Conceptualization:** Michael Toal.

**Data curation:** Michael Toal.

**Formal analysis:** Michael Toal, Clare McKeaveney.

**Funding acquisition:** Michael Toal.

**Investigation:** Michael Toal, Megan Raynor.

**Methodology:** Michael Toal, Clare McKeaveney.

**Project administration:** Michael Toal.

**Resources:** Michael Toal.

**Software:** Michael Toal.

**Supervision:** Ciaran O'Neill, Michael Quinn, Christopher Hill, Alexander Peter Maxwell.

**Writing – original draft:** Michael Toal.

**Writing – review & editing:** Ciaran O'Neill, Michael Quinn, Christopher Hill, Alexander Peter Maxwell.

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
