## [Decision Letter · Decision Letter 0]

10 Jul 2024

PONE-D-23-42064"Was my kidney biopsy worth it?" A qualitative phenomenological study of patient experiences and perceived barriers to kidney biopsyPLOS ONE

Dear Dr. Toal,

Thank you for submitting your manuscript to PLOS ONE. After careful consideration, we feel that it has merit but does not fully meet PLOS ONE’s publication criteria as it currently stands. Therefore, we invite you to submit a revised version of the manuscript that addresses the points raised during the review process.

We look forward to receiving your revised manuscript.

Kind regards,

Ankur Shah

Academic Editor

PLOS ONE

“MT is supported by a research award from the Northern Ireland Kidney Research Fund. This organisation was not involved in the design or conduct of this study.

https://nikidneyresearch.org/”

3. In the online submission form you indicate that your data is not available for proprietary reasons and have provided a contact point for accessing this data. Please note that your current contact point is a co-author on this manuscript. According to our Data Policy, the contact point must not be an author on the manuscript and must be an institutional contact, ideally not an individual. Please revise your data statement to a non-author institutional point of contact, such as a data access or ethics committee, and send this to us via return email. Please also include contact information for the third party organization, and please include the full citation of where the data can be found.

Reviewers' comments:

Reviewer's Responses to Questions

**Comments to the Author**

1. Is the manuscript technically sound, and do the data support the conclusions?

Reviewer #1: Yes

Reviewer #2: Yes

Reviewer #3: Yes

2. Has the statistical analysis been performed appropriately and rigorously? 

Reviewer #1: I Don't Know

Reviewer #2: Yes

Reviewer #3: N/A

3. Have the authors made all data underlying the findings in their manuscript fully available?

Reviewer #1: No

Reviewer #2: Yes

Reviewer #3: Yes

4. Is the manuscript presented in an intelligible fashion and written in standard English?

Reviewer #1: Yes

Reviewer #2: Yes

Reviewer #3: Yes

5. Review Comments to the Author

Reviewer #1: This study ist about the the physiological (pain of the procedure...) and psychological aspects (stress, anxiety and felt impact of the treatment course...) of kidney biopsy. The kidney biopsy is an invasive, important diagnostic methode in the nephrology. The study is retrospective and includes 10 partecipants, wich undergoes 23 kidney biopsies. There were used questionnaires. There were included patients with CKD and also kidney transplanted patients.

Up to now there are existing only few studies about the physic and psychological impact in patients. This is a very interestning and import argument. Very simple and clear the introducion and objectives of the study. Methode is simple.

The discussion could be more precise:

- Interestning is for example the impact of the age: how is the resilience in elder patients, have elder patients more pain? How do you explain the results

- It is also interestning to investigate the results from patients which the native kidney was biopsied versus these which the transplanted kidney was biopsied. Differed the answers of the questionnaires in the groups? How do you explained?

- It is also interestning to specified in the table the motivation of kidney biopsy: Acute kidney injury (AKI); nephrotic syndrome; autoimmune systemic disease with suspect renal manifestation? How differ the answers in the questionnaire?

- Many patients ondergoes multiple kidney biopsy: How is their resilience, pain, felt impact in treatment course? Are differences in confront to patients which ungergoes only one kidney biopsy?

Reviewer #2: This is an interesting study by Toal et al describing the patient perspective of undergoing a kidney biopsy using a phenomenological approach.

Overall, the methodology is well described and the core elements of the reporting checklist have been adhered to, which are strengths. I appreciate the context that this is a phenomenological approach, which has some strengths and weaknesses. I believe this would be a meaningful contribution to the literature, but I also have some concerns that I hope the authors could address to strengthen the manuscript.

Major concerns:

1) The context for each theme feels limited, and the representative quotes do not appear to fully justify some components of the Discussion. For example, the authors note in the Discussion section that "the individual who reported the most severe pain also described a significant psychological burden resulting from their kidney disease." However, the quotes in the Results section do not sufficiently describe this psychological burden. It would be helpful to add a few more quotes to the results section, perhaps in a Table or in the main body of the paper. This could help add more context for the general reader for each theme and subtheme outlined in the paper.

2) I struggled with some aspects of the discussion and results. In particular, I felt that the purpose of the paper was to describe the patient perspective of undergoing a kidney biopsy. However, there were a few sections that appeared to focus more on the experience of living with kidney failure, and were not necessarily related to receiving a kidney biopsy (for example, the notion of having cramps after football, or losing the ability to play rugby). It would be helpful to either clarify this differentiation more explicitly, or mention in the limitations section that some of the experiences described were related to kidney failure and not necessarily to kidney transplantation.

3) To help the general reader understand the methodology better, please provide more description as to how the qualitative findings were independently verified by an experienced researcher. For example, did the first author and this experienced researcher meet several times to go over the themes and come to consensus? Did the experience qualitative researcher provide asynchronous feedback by email? Did new themes or concepts emerge from these discussions? A fuller description of this approach could help reduce concerns of the undue influence of Dr. Toal's role as a nephrologist who performs kidney biopsy (on a side note, I acknowledge and appreciate that Dr. Toal went to great lengths to describe his position, his own potential biases, and his reflexivity in the methodology section of the paper!)

4) The authors go to great detail to discuss and differential whether biopsies were performed on native or transplant kidneys. However, I did not notice any discussion about whether patient perspectives were similar or different when they received a native or transplant biopsy. From a grounded theory approach, one would assume that these perspectives would be very different. I appreciate that this nuance may not emerge from a phenomenological approach, however, if it did emerge, it would be worth noting. If it did not emerge, please consider describing this in the limitations.

5) Similarly, please provide justification for why the authors chose to use a phenomenological approach over a grounded theory approach. It is reasonable to pursue a phenomenological approach here, but discussing the rationale for choosing this approach would help strengthen the paper.

Minor comments:

1) In the background section, please provide more information on the incidence of kidney biopsies. If I were to just read the paper without any other clinical context, I'm not sure I would understand how big of a problem the authors are trying to tackle.

2) The authors discuss that the findings may have limited generalizability. As this is a qualitative study, please consider discussing transferability instead of generalizability.

3) The authors discuss how their work could help inform equitable access to care. However, it was not clear to me how the authors could link their work to health equity. I did not see a clear delineation of disparities in kidney biopsies, or how this patient perspective could address disparities. I do note that the interview guide asked for ethnicity of participants, however the body of the paper des not mention ethnicity. Please consider including ethnicity data. Please also consider adding a stronger rationale for how the study could potentially improve equitable access, or consider removing this section altogether.

Reviewer #3: Toal et all carried out structure interviews with 11 patients who had undergone 23 kidney biopsies to get a sense of the themes of patient experience. The selection process was purposive, and the sample size, so no way to see if experience of participants who underwent native biopsy different from those who underwent transplant biopsies. A couple issues should be addressed.

1. Authors note the importance of knowing the person who performed the biopsy. But they note that MT as a physician may have influenced their results. How many of the biopsies did MT perform and how might that have influenced the interviews?

2. The psychological effects of the illness don’t seem a direct consequence of the biopsy but a consequence of the diagnosis that grew out of the biopsy. How do the authors tie this in with the biopsy.

3. Can the authors speculate if there would be a difference between those who underwent native kidney biopsies and those who underwent allograft biopsies. Native biopsy patients have no idea what’s happening unless the operator tells them in advance since they’re lying on their stomach.

4. Did any of the 10 suffered untoward consequences like bleeding, hematuria, hematoma after biopsy? How might the experience of those who experienced a complication differ from those who did not? If no one in the study group suffered a complication how generalizable are the findings to those patients?

6. PLOS authors have the option to publish the peer review history of their article (what does this mean?). If published, this will include your full peer review and any attached files.

Reviewer #1: No

Reviewer #2: No

Reviewer #3: **Yes: **George Bayliss, MD

---

## [Author Response · Author response to Decision Letter 0]

9 Aug 2024

5. Review Comments to the Author

Reviewer #1: This study ist about the the physiological (pain of the procedure...) and psychological aspects (stress, anxiety and felt impact of the treatment course...) of kidney biopsy. The kidney biopsy is an invasive, important diagnostic methode in the nephrology. The study is retrospective and includes 10 partecipants, wich undergoes 23 kidney biopsies. There were used questionnaires. There were included patients with CKD and also kidney transplanted patients.

Up to now there are existing only few studies about the physic and psychological impact in patients. This is a very interestning and import argument. Very simple and clear the introducion and objectives of the study. Methode is simple.

The discussion could be more precise:

Q1A- Interestning is for example the impact of the age: how is the resilience in elder patients, have elder patients more pain? How do you explain the results

A1A: Thank you for raising this point. In our small sample there did not appear to be any clear relationship between age and pain perception. The participant who was youngest at the time of biopsy (14 years) reported minimal pain, as did the oldest participants at time of biopsy (47 years). Although there was no clear correlation between age and perceptions of pain in this study, further research with larger cohorts would be required to try and answer this question. 

Q1B- It is also interestning to investigate the results from patients which the native kidney was biopsied versus these which the transplanted kidney was biopsied. Differed the answers of the questionnaires in the groups? How do you explained?

A1B: This is an important point and has also been raised by the other reviewers. Participants who had undergone a kidney transplant before their first biopsy appeared to better tolerate the procedure. There may be several reasons for this relating to the positioning of the patient, their previous experience of invasive procedures in the form of transplant surgery and familiarity with the clinical environment and staff. However this study did not control for these factors, therefore a cautious approach is required in interpretation, as this is beyond the scope of what can be determined by this qualitative approach. 

-Q1C It is also interestning to specified in the table the motivation of kidney biopsy: Acute kidney injury (AKI); nephrotic syndrome; autoimmune systemic disease with suspect renal manifestation? How differ the answers in the questionnaire?

A1C: As you rightly point out, there are many situations which may warrant a kidney biopsy, which may precipitate differing experiences for patients. The researchers did not have access to the medical records of participants; therefore these details are incomplete. Participants did volunteer their journey to a kidney biopsy in their own words, which provides some insight into the indications. In this study, several individuals had significant symptoms at the time of the biopsy, such as haemoptysis and leg oedema. Some individuals who were already symptomatic appeared to tolerate the procedure better than asymptomatic persons, however this was not consistent across the entire cohort.

Q1D- Many patients ondergoes multiple kidney biopsy: How is their resilience, pain, felt impact in treatment course? Are differences in confront to patients which ungergoes only one kidney biopsy?

A1D: In this study, 8 out of 10 participants underwent multiple kidney biopsies. In most cases, participants reported that the subsequent biopsies were better tolerated than their first biopsy. However, concluding that subsequent biopsies are better tolerated than the first is beyond the remit of qualitative research. 

We would suggest that one reason for this inference could be that participants knew what to expect from the procedure, rather than fearing the unknown. Another important issue was that participants who were undergoing a repeat kidney biopsy were aware of the long clinical observation time and brought resources with them to pass the time, minimising the monotony associated with a long observation period.

Reviewer #2: This is an interesting study by Toal et al describing the patient perspective of undergoing a kidney biopsy using a phenomenological approach.

Overall, the methodology is well described and the core elements of the reporting checklist have been adhered to, which are strengths. I appreciate the context that this is a phenomenological approach, which has some strengths and weaknesses. I believe this would be a meaningful contribution to the literature, but I also have some concerns that I hope the authors could address to strengthen the manuscript.

Major concerns:

Q2A: 1) The context for each theme feels limited, and the representative quotes do not appear to fully justify some components of the Discussion. For example, the authors note in the Discussion section that "the individual who reported the most severe pain also described a significant psychological burden resulting from their kidney disease." However, the quotes in the Results section do not sufficiently describe this psychological burden. It would be helpful to add a few more quotes to the results section, perhaps in a Table or in the main body of the paper. This could help add more context for the general reader for each theme and subtheme outlined in the paper.

A2A: Thank you for raising this point. We have added additional participant quotes to help expand these themes. These additional quotes provide more insights into areas detailed in the discussion. Upon deliberation within the research team, we concluded that it was not constructive to highlight the psychological distress of just one individual, as all participants describe varying degrees of a physical and emotional burden relating to their biopsy and related kidney disease. Therefore this sentence has been removed from the discussion section.

Q2B: 2) I struggled with some aspects of the discussion and results. In particular, I felt that the purpose of the paper was to describe the patient perspective of undergoing a kidney biopsy. However, there were a few sections that appeared to focus more on the experience of living with kidney failure, and were not necessarily related to receiving a kidney biopsy (for example, the notion of having cramps after football, or losing the ability to play rugby). It would be helpful to either clarify this differentiation more explicitly, or mention in the limitations section that some of the experiences described were related to kidney failure and not necessarily to kidney transplantation.

A2B: The reviewer is correct that the main purpose of the paper was to describe the patient perspective of undergoing a kidney biopsy. This is a key issue that has not been well described previously in the literature. The focus of the study was to try and understand the lived experience of having a kidney biopsy, however many participants took this opportunity to also describe, in great detail, their experience of living with kidney disease. Although this is not always directly related to the biopsy, this procedure occurred within the context of their illness (kidney disease), therefore the two are inevitably intertwined through the patient journey. We felt it was important to depict the mental health context in which patients are asked to undergo this invasive procedure, as these individuals’ vulnerability is further tested on top of the physical and emotional toll of their illness. This point has been expanded further in the discussion section.

Q2C: 3) To help the general reader understand the methodology better, please provide more description as to how the qualitative findings were independently verified by an experienced researcher. For example, did the first author and this experienced researcher meet several times to go over the themes and come to consensus? Did the experience qualitative researcher provide asynchronous feedback by email? Did new themes or concepts emerge from these discussions? A fuller description of this approach could help reduce concerns of the undue influence of Dr. Toal's role as a nephrologist who performs kidney biopsy (on a side note, I acknowledge and appreciate that Dr. Toal went to great lengths to describe his position, his own potential biases, and his reflexivity in the methodology section of the paper!)

A2C: This is an important methodological issue, thank you for raising this. We have added further detail within the manuscript of how the stepwise method for Colaizzi’s phenomenological approach was utilised. The first author (MT) identified codes independently in the first instance. Subsequently he met with the experienced qualitative researcher (CMcK) to outline the purpose of the study and relevant methodologies. She then read two sample transcripts, which were the most comprehensive and contrasting and coded these independently, blinded to the initial codes. These new themes were then discussed iteratively to derive new codes, and then expanded to the other transcripts. Each draft of the manuscript was read and amended by CMcK to maintain consistent coding across all participants.

Q2D: 4) The authors go to great detail to discuss and differential whether biopsies were performed on native or transplant kidneys. However, I did not notice any discussion about whether patient perspectives were similar or different when they received a native or transplant biopsy. From a grounded theory approach, one would assume that these perspectives would be very different. I appreciate that this nuance may not emerge from a phenomenological approach, however, if it did emerge, it would be worth noting. If it did not emerge, please consider describing this in the limitations.

A2D: This qualitative study did not control for distinctions between the lived experience of individuals who undergo native and transplant biopsies, therefore a cautious approach is required in interpretation. Utilising a phenomenological approach in this small cohort, the lived experience of participants who had been previously transplanted at the time of their first biopsy may have been different to those who had not. There may be several reasons for this relating to the positioning of the patient, their previous experience of invasive procedures in the form of transplant surgery and familiarity with the clinical environment and staff. However further research would be needed to draw firm conclusions in this area.

Q2E: 5) Similarly, please provide justification for why the authors chose to use a phenomenological approach over a grounded theory approach. It is reasonable to pursue a phenomenological approach here, but discussing the rationale for choosing this approach would help strengthen the paper.

A2E: We agree with the reviewer that there are alternate qualitative research approaches to try and answer research questions.

A phenomenological approach was used to understand the direct ‘lived experiences’ of the patients undergoing a kidney biopsy i.e. we were interested in their own subjective understanding of having a kidney biopsy. There is a paucity of relevant literature in this area. If there were prohibiting circumstances that could prevent patients coming forward for biopsy, this could act as a barrier to timely diagnosis and treatment. 

This explanation has been expanded in the manuscript for a more thorough justification of the methodological decision. 

Minor comments:

Q2F: 1) In the background section, please provide more information on the incidence of kidney biopsies. If I were to just read the paper without any other clinical context, I'm not sure I would understand how big of a problem the authors are trying to tackle.

A2F: We agree entirely that this would help to provide context for the paper. Details of biopsy numbers within one region of the UK and internationally in terms of rates per million population annually have been added in the Introduction to the manuscript.

Q2G: 2) The authors discuss that the findings may have limited generalizability. As this is a qualitative study, please consider discussing transferability instead of generalizability.

A2G: Thank you for spotting this error. We have changed this to transferability.

Q2H: 3) The authors discuss how their work could help inform equitable access to care. However, it was not clear to me how the authors could link their work to health equity. I did not see a clear delineation of disparities in kidney biopsies, or how this patient perspective could address disparities. I do note that the interview guide asked for ethnicity of participants, however the body of the paper des not mention ethnicity. Please consider including ethnicity data. Please also consider adding a stronger rationale for how the study could potentially improve equitable access, or consider removing this section altogether.

A2H: This is an important point you have raised. Data on ethnicity has now been added. In terms of access to care, the authors do not propose that certain groups of individuals defined by age, sex or race are systematically disadvantaged with poorer access to kidney biopsy. We sought to examine the origins of patient reluctance or hesitation when this investigation was recommended by their healthcare provider. Specific patient-related factors may limit uptake, such as childcare provision to mothers of young children or loss of income for self-employed individuals. In our study, women reported specific challenges relating to urination whilst on bedrest during the observation period, which is a potential disparity that could be highlighted in advance for female patients. Understanding which patient-related barriers contribute to biopsy hesitancy allows for healthcare professionals to adapt services to help ensure individuals who are deemed to require this procedure can access it.

Reviewer #3: Toal et all carried out structure interviews with 11 patients who had undergone 23 kidney biopsies to get a sense of the themes of patient experience. The selection process was purposive, and the sample size, so no way to see if experience of participants who underwent native biopsy different from those who underwent transplant biopsies. A couple issues should be addressed.

Q3A: 1. Authors note the importance of knowing the person who performed the biopsy. But they note that MT as a physician may have influenced their results. How many of the biopsies did MT perform and how might that have influenced the interviews?

A3A: MT was not involved in the direct clinical care of any of the individuals who participated in the study at any point. Therefore, he performed none of the biopsies on these patients. MT has performed approximately 50 kidney biopsies across a five-year career in nephrology at the time of interview, however none of the patients he had biopsied were included in this study.

Q3B: 2. The psychological effects of the illness don’t seem a direct consequence of the biopsy but a consequence of the diagnosis that grew out of the biopsy. How do the authors tie this in with the biopsy.

A3B: This is an important distinction and has been highlighted by other reviewers. The kidney biopsy occurred within the context of each individual’s illness and the associated physical and emotional toll. We felt it was important to provide context to the reader that this invasive procedure was undertaken at a time when the participant was already vulnerable. The psychological effects may not have always been directly related to the procedure; however the biopsy was undertaken during the patient journey of living with their illness, therefore we felt it was important to allow participants to explain what this meant to them within that framework.

Q3C: 3. Can the authors speculate if there would be a difference between those who underwent native kidney biopsies and those who underwent allograft biopsies. Native biopsy patients hav

---

## [Decision Letter · Decision Letter 1]

30 Aug 2024

'Was my kidney biopsy worth it?' - A qualitative phenomenological study of patient experiences and perceived barriers to kidney biopsy

PONE-D-23-42064R1

Dear Dr. Toal,

We’re pleased to inform you that your manuscript has been judged scientifically suitable for publication and will be formally accepted for publication once it meets all outstanding technical requirements.

Kind regards,

Ankur Shah

Academic Editor

PLOS ONE

Additional Editor Comments (optional):

Reviewers' comments:

Reviewer's Responses to Questions

**Comments to the Author**

1. If the authors have adequately addressed your comments raised in a previous round of review and you feel that this manuscript is now acceptable for publication, you may indicate that here to bypass the “Comments to the Author” section, enter your conflict of interest statement in the “Confidential to Editor” section, and submit your "Accept" recommendation.

Reviewer #1: All comments have been addressed

Reviewer #3: All comments have been addressed

2. Is the manuscript technically sound, and do the data support the conclusions?

Reviewer #1: Yes

Reviewer #3: Yes

3. Has the statistical analysis been performed appropriately and rigorously? 

Reviewer #1: N/A

Reviewer #3: Yes

4. Have the authors made all data underlying the findings in their manuscript fully available?

Reviewer #1: Yes

Reviewer #3: Yes

5. Is the manuscript presented in an intelligible fashion and written in standard English?

Reviewer #1: Yes

Reviewer #3: Yes

6. Review Comments to the Author

Reviewer #1: Manuscript Number: PONE-D-23-42064R1

Manuscript Title: 'Was my kidney biopsy worth it?' - A qualitative phenomenological study of patient experiences and perceived barriers to kidney biopsy

The suggestions for improvement have been well implemented in the revised version.

The number of kidney biopsies in Scotland and international was added. Ethnicities were included in the description of the collectives.

The proposed aspects were addressed in the discussion: It is clear to the reader that no conclusions can be drawn from this data regarding interpretation in terms of age and gender as the collective is limited in number. It is stated that both second biopsies and biopsies in kidney transplant patients are better tolerated. Probable reasons are also given: Previous experience with invasive investigations results in higher tolerance.

Reviewer #3: The authors have addressed my concerns. Many of my questions couldn't be addressed by the study as designed. So be it. It's still an interesting concept to look at the biopsy from the patient's perspective and in the perspective the illness. Several participants had native biopsies followed by biopsies. I'd love to know how their biopsy changed from a diagnostic procedure as they lost kidney function to one post-transplant. One other question that might be relevant is whether the transplant biopsies were done for cause or whether they were protocol biopsies for surveillance.

7. PLOS authors have the option to publish the peer review history of their article (what does this mean?). If published, this will include your full peer review and any attached files.

Reviewer #1: No

Reviewer #3: **Yes: **George Bayliss, MD

---

## [Editor Report · Acceptance letter]

3 Sep 2024

PONE-D-23-42064R1 

PLOS ONE

Dear Dr. Toal, 

I'm pleased to inform you that your manuscript has been deemed suitable for publication in PLOS ONE. Congratulations! Your manuscript is now being handed over to our production team.

Kind regards, 

on behalf of

Dr. Ankur Shah 

Academic Editor

PLOS ONE